# Mechanistic Insight into the Early Stages of Toroidal Pore Formation by the Antimicrobial Peptide Smp24

**DOI:** 10.3390/pharmaceutics15102399

**Published:** 2023-09-28

**Authors:** Magnus Bertelsen, Melissa M. Lacey, Tim Nichol, Keith Miller

**Affiliations:** Biomolecular Sciences Research Centre, Sheffield Hallam University, Sheffield S1 1WB, UK

**Keywords:** antimicrobial peptides, molecular dynamics simulations, mechanism of action, membrane pore, patch-clamp electrophysiology, disordered toroidal pore, early-stage pore formation

## Abstract

The antimicrobial peptide Smp24, originally derived from the venom of *Scorpio maurus palmatus*, is a promising candidate for further drug development. However, before doing so, greater insight into the mechanism of action is needed to construct a reliable structure–activity relationship. The aim of this study was to specifically investigate the critical early stages of peptide-induced membrane disruption. Single-channel current traces were obtained via planar patch-clamp electrophysiology, with multiple types of pore-forming events observed, unlike those expected from the traditional, more rigid mechanistic models. To better understand the molecular-level structures of the peptide-pore assemblies underlying these observed conductance events, molecular dynamics simulations were used to investigate the peptide structure and orientation both before and during pore formation. The transition of the peptides to transmembrane-like states within disordered toroidal pores occurred due to a peptide-induced bilayer-leaflet asymmetry, explaining why pore stabilization does not always follow pore nucleation in the experimental observations. To fully grasp the structure–activity relationship of antimicrobial peptides, a more nuanced view of the complex and dynamic mechanistic behaviour must be adopted.

## 1. Introduction

The increasing levels of antimicrobial resistance of many pathogenic microorganisms have led to a crucial need for the development of new alternative antimicrobial drugs with novel mechanisms of action [1]. Antimicrobial peptides (AMPs) have been highlighted multiple times as a class of drug that could play a key role in tackling the increasing difficulties encountered in treating microbial infections [1,2]. With a unique mechanism of action targeting the cell membrane as the main point of attack, in addition to a variety of other potential targets, resistance development towards AMPs is less prevalent compared to traditional antibiotics [3]. While AMPs are a rich and diverse group of compounds, they are most commonly amphiphilic and alpha-helical in structure with an overall positive charge. Their main mechanism of action involves the disruption of the bacterial cell membrane(s), causing leakage of the internal cellular components and eventual cell death. However, the specifics of the mechanism behind the disruption are still highly debated. While the three traditional mechanisms of actions (barrel-stave, toroidal pore and carpet mechanism) are often still described [4], new alternative mechanisms are being proposed focusing less on the AMPs acting like ion channels by forming distinct pores and more on them acting as bio-detergents in a dynamic relationship with the bacterial membrane(s) [5,6].

Venoms have been found to be a rich reservoir of naturally occurring AMPs, with one example being the venom of the Egyptian scorpion *Scorpio maurus palmatus*. Multiple AMPs have been found in this venom, including the 24-residue peptide Smp24 (IWSFLIKAATKLLPSLFGGGKKDS). This peptide has previously been shown to have broad antimicrobial activity against a range of clinically relevant bacteria species [7,8]. While unstructured in water, the peptide adopts a majority helical structure in a 60% trifluoroethanol solution, which is typical of amphiphilic AMPs [7]. Interestingly, when comparing the primary sequence of Smp24 with similar AMPs, another possible structural motif can be identified. Like Smp24, melittin [9], magainin 2 [10], brevinin-1EMa [11], piscidin 1+3 [12] and gaegurin P14 [13] all have a proline or glycine residue positioned around the middle of the sequence and cause a kink in the helical region that is also likely present in Smp24 due to a proline residue at position 14 [7]. The mechanism of action of Smp24 has previously been investigated using several different biophysical techniques such as atomic force microscopy (AFM), quartz crystal microbalance with dissipation monitoring (QCM-D) and liposomal leakage assays showing a clear dependency on the specific lipid composition [14]. Further study around how the peptide interacts with the membranes on a molecular level could help in establishing an improved structure–activity relationship allowing the effective employment of a rational structure-based drug design strategy.

Advances in planar patch-clamp equipment such as the port-a-patch by Nanion (Munich, Germany) allows for improved in vitro investigation of peptide-induced pore formation at the level of a single pore [15,16]. By measuring the peptide-induced change in conductance across a bilayer, information can be obtained related to the molecular-level structure and behaviour of the pores. However, to fully understand the structure mechanism relationship of the peptides, knowledge about the pore structure by itself is not enough. The structure of the peptides themselves, and how this structure allows them to take part in and stabilise the overall structure of the pores, must also be considered to allow for better design of new peptides in the future.

Molecular dynamics (MD) simulations have been shown to be an extremely useful tool for in silico investigations of the biophysical properties of antimicrobial peptides at a molecular level [17,18,19]. MD simulations have previously been used to explore several different individual stages of the mechanism of action of AMPs, such as their mechanism of insertion into lipid bilayers [20,21], their position and interactions once they are fully inserted into the bilayer [12,19], and their configuration when associated with a membrane pore structure [22,23,24]. 

In this study, we have utilised planar patch-clamp electrophysiology to investigate the early stages of peptide-induced membrane disruption and pore formation caused by Smp24 and contextualised these observations using MD simulations, to better explain the underlying biophysical phenomena leading to the membrane disruption and how the specific structure of the peptides relates to the stabilization of the peptide-pore assemblies. 

## 2. Materials and Methods

All reagents and materials were purchased from Sigma-Aldrich (Gillingham, UK) unless otherwise stated.

### 2.1. Patch Clamp

Giant unilamellar vesicles (GUVs) for the patch-clamp experiments were produced using the vesicle prep pro from Nanion (Munich, Germany). A phospholipid mixture 1,2-dioleoyl-sn-glycero-3-phosphocholine (DOPC) and 1,2-dioleoyl-sn-glycero-3-phospho-(1′-rac-glycerol) (DOPG) (3 mg/mL 1:1 ratio) in chloroform was placed on an ITO slide and allowed to fully dry. Electroformation was performed in the presence of a 1 M sorbitol solution using a stepwise protocol. At 37 °C and an amplitude of 5 Hz, the voltage was raised to 3 V over 5 min and then held there for 200 min. The voltage was then lowered to 0 over 3 min and the GUVs were harvested and stored at 4 °C and used within 4 days.

Electrophysiology experiments were performed on planar lipid bilayers using Nanion’s (Munich, Germany) Port-a-Patch planar patch-clamp setup. The bilayers were formed at pH 4 (200 mM KCl, 10 mM HEPES) on a 3–5 MOhm borosilicate chip (Nanion, Munich, Germany) by applying 10–30 mbar negative pressure till a GOhm seal was achieved. The buffer was exchanged to pH 7 (200 mM KCl, 10 mM HEPES) and the current was measured at a holding potential of +60 mV for five minutes to insure the stability of the bilayer. Finally, different concentrations of Smp24 diluted in the pH 7 buffer were added (active concentrations of 2.29–7.76 µM) and the current trace was recorded using an HEKA EPC 10 amplifier (Heka Elektronik, Lambrecht, Germany), with 5 repeats for each peptide concentration.

### 2.2. Molecular Dynamics Simulations

To investigate the structure of Smp24 and its interplay with a lipid bilayer through different stages of the mechanism of action, three stages of MD simulations were performed. All simulations were performed using the Gromacs 2020.2-4 packages, using the leapfrog algorithm with a 2 fs timestep, hydrogen bond constraints were incorporated using the Lincs algorithm; Wan der Vaals and short-range electrostatic interactions cut-offs were 1.2 nm, a Nosé–Hoover thermostat was used for temperature control and the Parrinello-Rahman barostat for pressure control with semi-isotropic conditions. Unless otherwise specified, all production runs were performed in the NPT ensemble at 293 K, 1 atm. The starting 3D structure of Smp24 used in all models was generated using the Pepfold3 server [25]. The simulations were analysed using VMD [26] and Gromacs [27] with the FATSLiM addon [28].

#### 2.2.1. Single Peptide Bilayer Models

To investigate the basic structure of the bilayer-associated peptide, a simple single-peptide bilayer model was designed. The base bilayer model was built using CHARMM-GUI [29] with both the lipid and peptide topology being described using the CHARMM36m force field. The bilayer was made using 72 DOPG and 72 DOPC lipids (approximately 7 nm^2^ in size). A 3 nm TIP3P water layer was added to each side of the bilayer, with a single peptide being placed in the centre of the xy plane, approximately 1.5 nm above the bilayer, with the helix paralleled with the bilayer. Potassium or chloride ions were added to neutralise the system. Following an energy minimisation, the models were equilibrated in 3 steps. Steps 1 and 2 were 100 ps in the NVT and NPT assembly, respectively, followed by a longer 900 ps NPT simulation. Positional restraints were applied to the peptide in all steps and to the lipids in steps 1 and 2. Following the equilibration, 3 replica production simulations were performed, ranging from 500 to 1000 ns in length, to account for the variation in the time taken for the peptide to fully insert within the bilayer.

#### 2.2.2. Multi-Peptide Bilayer Models

To investigate the effects of peptide insertion on the properties of the bilayer, several multi-peptide bilayer models were made. The starting points for these models were created by performing single-peptide bilayer simulations as described above except smaller 3.3 nm^2^ or 4 nm^2^ bilayers were used. Once full peptide insertion was achieved, the models were multiplied in a 4 × 4 or 3 × 3 matrix in the x and y directions using the gmx genconf function using a similar approach as Chen et al. [30]. This approach allows for the rapid production of bilayer models with several inserted peptides and ensures that all of them are inserted in the same bilayer leaflet, but leaves the peptides at an artificial, symmetric distance from each other. Therefore, prior to analysis, the larger models were first equilibrated by simulating at an elevated temperature (323 K) for 500 ns to facilitate rapid diffusion and mixing of the peptides in the bilayer plane. Lastly, models were simulated for 250 ns at the normal temperature (293 K), with the trajectories from this simulation used for analysis.

#### 2.2.3. Peptide-Pore Models

To investigate the structure, position and orientation of Smp24 while associated with a pore, toroidal pores were induced into the endpoint of the previously described peptide-membrane models via electroporation. This was performed by applying a 0.3 V/nm electric field tangential to the bilayer. After a consistent pore had formed, the strength of the electric field was lowered to 0.065 V/nm and the x-y compression was set to 0. This allowed for the modelling of a stable pore using both a 7 nm^2^ bilayer with a single peptide inserted (3 repeats, 500 ns in length) and a 14.2 nm^2^ bilayer with 16 peptides inserted (5 repeats, 100 ns in length). A more detailed protocol can be found in Appendix A.

#### 2.2.4. Statistical Analysis

The patch clamp-derived pore formation kinetics for Smp24 were analysed using unpaired two-sided *t*-tests with independent spread. An α value of 0.05 was used for identification of significant differences.

## 3. Results

### 3.1. Evaluation of Peptide-Induced Membrane Disruption Using Patch-Clamp Electrophysiology

To investigate the pore formation of Smp24, planar patch clamp using 1:1 DOPC:DOPG bilayers was performed at five different peptide concentrations, with five repeats of each concentration. The bilayer composition was chosen to broadly mimic the negative charge of the bacterial (inner) membrane [31], while the unsaturated lipid chains allowed the bilayer to remain in the liquid phase state at room temperature. Due to the broad spectrum of activity of Smp24 [7], further customisation of the lipid composition to mimic a specific bacteria species was not deemed necessary to achieve a reasonable degree of representativity, at least as to what could be expected from a pure synthetic membrane model.

The kinetic aspects of the pore formation were analysed based on the time it took for the first conductance event to occur and the time it took between the peptide addition and the complete destruction of the bilayer. In 64% of the experimental runs, the lag time between the peptide addition and the occurrence of the first conductance event was 20 s or less. However, at peptide concentrations below 4.85 µM, longer lag periods (1–26 min) start to occur, although not consistently for all repeats (Figure 1A).

Similarly, highly variable lag times between peptide addition and the complete destruction of the bilayers was also seen. At high peptide concentrations, the bilayers were often destroyed within the first 5 min of peptide addition, whereas at concentrations below 3.9 µM, a large increase in the variation of destruction time was seen, and the average kinetics were slowed (Figure 1B). Furthermore, for 40% of the experimental runs at the two lowest peptide concentrations (2.91–3.88 µM), the bilayer was still intact after 30 min, with half of those experiments resulting in no membrane disruption at all.

To further evaluate the individual membrane disruptive events induced by the Smp24 peptide, the current traces from the individual experiments were qualitatively analysed. Multiple different types of event signatures were found throughout most runs, which could broadly be categorized into three distinct event types, multilevel, spike and erratic, based on Chui et al. [32].

Multilevel events most closely represent what would be expected for a distinct pore. The start and end of the signal are both sharp transitions from the baseline. During the event, a consistent but highly variable increase in conductance could be observed, lasting from 100 ms to around 3 s (Figure 2A). For longer events of this type, an average conductance level could be estimated using amplitude histograms fitted with a Gaussian distribution function. However, the spread of the current distribution corresponding to the multilevel event was much greater than the baseline. Comparing average current measured between different multilevel events also showed a large distribution of values ranging from 2.8 to 18.3 pA.

Like multilevel events, spike events (Figure 2B) also have a clear transition to and from the baseline, albeit the length of the event is much shorter (<50 ms). Again, no consistent average/maximum current level could be found when comparing different individual events, even within a singular run. Spike events were both observed as lone events or as multiples with a short time gap between them.

Erratic events (Figure 2C) were long (often multiple seconds) but with a relatively low and very variable conductance level. Unlike the other event types, they did not have a very clear beginning or end, but rather gradually increased or decreased the conductance. Over the lifetime of the event, the conductance level could shift multiple times and often return to a partial baseline with increased noise and conductance in between. Due to the more gradual conductance evolution, the amplitude histograms did not show an independent conductance level for the event, but rather a broadening of the baseline peak.

Multilevel, spike and erratic event types were found at all peptide concentrations; however, as with the kinetic data, a large variation in the number of events were seen within the same peptide concentration. Therefore, no consistent relationship between the peptide concentration and the likelihood of a specific event type occurring could be found. However, in general, across all the peptide concentrations, spike events were the most likely to occur followed by erratic events and lastly multilevel events.

### 3.2. Investigation of Interactions between Smp24 and Lipid Bilayers Using MD Simulation

In order to contextualize the different conductance event types on a molecular level, MD simulations of Smp24 in association with the lipid bilayer were performed. However, while the patch-clamp experiments can only provide information related to the peptide bilayer interactions after the formation of a pore has occurred, MD simulation can also predict the behaviours leading up to these events. To establish a baseline for the behaviour and structure of the peptide, simulations of the interactions between Smp24 and phospholipid bilayers were first performed, mimicking the conditions prior to pore formation.

#### 3.2.1. Insertion of Smp24 into Bilayers

In all simulations, the peptide followed a consistent mechanism of insertion into the negatively charged bilayer that can be separated into multiple distinct steps. Following an initial lag period with some sporadic electrostatic interactions between the peptide and the bilayer, the first consistent step seen in all the simulations is the anchoring of the N-terminal region to the bilayer (Figure 3B). This interaction is, initially, driven by electrostatic interactions between the N-terminal amine and the lipid phosphate groups then, following a short delay, supported by further electrostatic interactions between the bilayer and the two lysine residues (lys7 and lys11) positioned in the helical part of the peptide (Figure 3A). In this position/orientation, most of the hydrophobic residues were orientated facing away from the bilayer except for the sidechains of the N-terminal ile1 and phe4, so hydrophobic interactions were limited to those residues. Due to the position of the helical lysine residues, the helical region of the peptides was orientated with a tilted angle relative to the bilayer normal of 125–140 degrees which inhibits interactions between the latter half of the peptide and the bilayer (Figure 3C). The length of this stage of the insertion varied from a few ns to hundreds of ns, likely dependent on how consistent the lysine-phosphate interactions were. The next stage of the insertion is defined by a major rotation of the helical region of the peptide, changing the orientation of the hydrophobic residues to facing down towards the core of the bilayer (Figure 3D). This rotation also drove further changes to the peptide orientation, such as a reduction in the tilt angle and a change in the overall position of the peptide’s centre of mass, bringing it closer to the core of the bilayer. These processes were not instant, taking around 50 ns or more from the beginning of the rotation until the peptide reached a stable orientation and insertion depth (between 4.2 × 10^5^ ps and 5.5 × 10^5^ ps for the repeat shown in Figure 3).

#### 3.2.2. Structure, Orientation and Position of the Inserted Smp24

Following the full insertion of a single Smp24 peptide into a bilayer, a relatively consistent 3D structure was observed for the peptide in all repeat simulations (Figure 4A). For further analysis, this structure was separated into several structural/functional regions with distinct positional distributions relative to the bilayer leaflet (Figure 4B). The peptide adopted a helical structure between residues 1 and 17 which could be divided into two distinct regions separated by a kink induced by the proline residue at position 14. The primary helical region (r1–13) was the part of the peptide that was inserted the deepest into the bilayer with its position overlapping with the lipid glycerol esters and the top of the acyl chains. It was orientated in parallel with the bilayer surface with an average tilt of between 92 and 102 degrees. The secondary helical region (r14–17) was positioned at around the same level, although it varied somewhat within each simulation as the direction of the bend is flexible. Towards the C-terminal end, the peptide adopted a random coil structure which again could be separated into two regions. The last four residues are all polar or charged and can be combined as a tail region. These residues were positioned almost exclusively amongst the headgroups of the lipids, although with a high degree of positional flexibility, as seen with the broad partial density peak and much higher RMSF compared to the rest of the peptide (Figure 4C). The elevated positioning of the tail region is possible due to three glycine residues (r18–20) that serve as a linker region between the secondary helix and the tail.

#### 3.2.3. Concentration-Dependent Effects of Bilayer Properties

To evaluate if the peptide insertion affected the biophysical properties of the bilayer, simulations were performed at different peptide-to-lipid ratios.

The area per lipid and bilayer thickness was compared between a bilayer only simulation and three simulations with peptides inserted at different peptide-to-lipid ratios (Table 1). Both aspects seemed to be dependent on the peptide-to-lipid ratio. As the number of peptides increased, the area per lipid also increased while the bilayer thickness decreased.

The deuterium order parameters of the lipid chains (Scd) were also investigated to evaluate if the inserted peptides affected the lipid order (Appendix A). A small increase in the order could be seen for the top of the lipid chain but the main effect was a concentration-dependent reduction in the order for the lower half of the lipid chains between C13 and C18 (Table 1). No differences could be seen between the bilayer-only simulation and the lowest peptide-to-lipid ratio (less than 1% difference); however, as more peptides were inserted into the bilayer, the lipid order started to be affected (6.4 to 12.9% reduction in the Scd).

Additionally, the lateral diffusion constants for the peptides were also calculated for the simulations using the linear part of their mean square displacement. Again, a concentration-dependent effect could be seen, with the ability of the peptides to freely diffuse around the bilayer plane being inhibited as the number of inserted peptides increased.

#### 3.2.4. Modelling of the Pore-Associated Peptide Configurations

To investigate how Smp24 might associate with a membrane pore and which configurations the peptide could adopt in order to stabilise the pore structure, pores were created in bilayer models with Smp24 already inserted. This was carried out by applying a strong electric field across the bilayer, leading to pore formation via electroporation. Once a small pore had been created, the strength of the electric field was reduced which, together with locking the expansion of the bilayer in the x and y direction, resulted in a dynamic but stable pore, with a pore lumen diameter of around 3 nm. Three repeat simulations were performed, each with a different starting point for the pore position relative to the position of the single Smp24 peptide. Throughout these simulations, two different pore-associated peptide configurations were observed, where in both cases the peptide was positioned at the interface between the pore and the rest of the bilayer. However, at no point did the peptide insert fully into the pore lumen as would be expected with a true transmembrane peptide configuration.

In the first configuration (Figure 5A), the peptide was orientated with the N-terminal towards the centre of the pore. The primary helical region was positioned at pore–bilayer interface and thus, to follow the higher curvature of the bilayer in this region, it adopted a higher tilt angle (120 degrees relative to the bilayer normal) compared with the normal orientation of the helix in the non-pore bilayer. The secondary helix was positioned further away from the pore where the bilayer curvature was less extreme and thus adopted a lower tilt angle (100 degrees vs. bilayer normal) more in line with the normal orientation. The Trp2 residue was inserted the deepest within the pore, with an average position around 0.6 nm above the bilayer centre.

In the second configuration (Figure 5B), the helical regions of the peptide were positioned in reverse. As such, the secondary helical region was now orientated towards the centre of the pore. Therefore, it was the secondary helix that adopted the more tilted state (65 degrees vs. bilayer normal), while the primary helix was only tilted slightly (84 degrees vs. bilayer normal) compared with the normal orientation. In this configuration, the leu16 residue was positioned the deepest at an average position of around 0.85 nm above the bilayer centre.

In both cases, the tail region seemed to behave relatively independently of the membrane pore, while the helical regions facilitated the interactions between the peptide and the pore.

Throughout the three simulations (combined 1.5 µs simulation time), the peptide spent approximately 150 ns in the A configuration, 720 ns in the B configuration and 630 ns in a configuration not associated with the pore. The A configuration only occurred in the one simulation where the pore nucleation site was close to the N-terminal of the peptide whereas the peptide transitioned to the B configuration at some point in all three simulations.

As no transmembrane configuration was observed throughout the simulations, a centre of mass pull function was applied to artificially position the peptide deeper within the pore, to evaluate the feasibility of a more deeply inserted configuration. Starting from either of the pore-associated configurations, the peptide was pulled towards the centre of the pore in both the x and z direction. Three different configurations were generated from each starting point with the peptide inserted into the pore to increasing degrees. However, following 50 ns of simulation, the peptide had in all cases either returned to the original interface-associated configuration or moved completely away from the pore, thus indicating that a more deeply inserted state is not favourable under the applied conditions (Appendix A).

To investigate if a change in the peptide-to-lipid ratio and the presence of multiple peptides could affect the pore-associated configurations, pores were induced into the 16-peptide model described earlier. Similar electric field conditions as in the single peptide simulations were used. However, the larger size of the bilayer meant that the pore also grew to a larger size, with a diameter of around 5 nm.

As with the single peptide simulations, most of the peptides adopted either a configuration associated with the pore interface or one completely independent of the pore. The number of peptides associated with the pore at the endpoint of each simulation ranged from four to five (Figure 6F), with 33% being in the A configuration, 58% in the B configuration and 8% adopting a more sideways-facing configuration. However, while the peptide configurations in the multi-peptide setup broadly aligned with what was observed for the single peptide, in some cases the specific positions and orientations of individual peptides were more extreme. In all the simulations, one to three of the peptides adopted a more deeply inserted position with part of the peptide reaching below the centre of the bilayer (Figure 6A–E). Both types of pore-associated configurations were observed to be able to adapt to this greater inserted state. However, to accommodate the new positions of the peptides relative to the curvature of the bilayer, increases in the tilt angles were observed. The A configuration (Figure 6A,B) was not dissimilar to the single peptide configuration although the tilt angle increased by a further 20–30 degrees. However, for the B configuration (Figure 6C,D), the deeper insertion meant that the primary helix was now also placed within the pore lumen and thus had to adopt a much steeper tilt angle (40–70 degrees) in order to align with the bilayer curvature.

Even though these configurations are more extreme, the peptides were still mainly associated with the top half of the pore and thus a true transmembrane configuration was not observed.

To investigate how the increased peptide-to-lipid ratio affected the properties of the bilayer in the membrane pore simulations, the movement of lipids between the upper and lower leaflets was quantified. Due to the presence of the membrane pore, the lipids could relatively easily translocate between the leaflets without the need for a slow lipid flipflop process to occur. The one-directional electrical field applied in these simulations caused the negatively charged DOPG to move from the top leaflet to the bottom which can be seen occurring over time in the simulations with a low peptide-to-lipid ratio (Figure 7A). However, in these simulations, the translocation of the DOPG lipids was counterbalanced by an equal but opposite movement of the neutral DOPC lipids, leaving the lipid density of the bottom leaflet relatively constant over time. This was not the case for the simulations with a high peptide-to-lipid ratio. The demixing of the DOPG and DOPC lipids still occurred; however, an overall net movement of lipids away from the top leaflet, which had the peptide inserted, was observed, leading to an increased combined-lipid density in the bottom leaflet (Figure 7B).

## 4. Discussion

### 4.1. Experimental Investigation of the Mechanisms of Pore Formation of Smp24

The mechanism by which AMPs induce the formation and stabilisation of membrane pores is complex. Several different models have been proposed to explain the underlying molecular-level peptide-pore structures which facilitate membrane disruption, the main mechanism by which antimicrobial peptides exert their antimicrobial properties. As previously mentioned, the three traditional models (barrel stave, toroidal pore and carpet) are the models most frequently described, although many others have been proposed [4]. Previously, the pore formation induced by Smp24 was investigated using atomic force microscopy (AFM) showing the formation of stable, differently sized circular pores with an average diameter of 80 ± 40 nm after 30 min of incubation [14]. The presence of distinct pores of greatly varying sizes indicates that, at least under these conditions, the pore structure most closely aligns with toroidal pore model [14]. However, the structure of these large mature pores does not necessarily correlate with the structure at the beginning of the pore formation process. Investigating the pore structure during these early stages of the disruption process may make it easier to correlate the structure of the individual peptide with the mechanism of pore formation and stabilisation, as fewer peptides likely take part in the peptide-pore assembly.

Single-channel-level patch-clamp investigations of peptide-induced pore formation enable the measurement of the increase in conductance across a membrane due to this membrane disruption. These current spikes are directly correlated to molecular-level changes to the structure of the bilayer and can therefore give insight into the transient structure of these pores in the early stages of the membrane disruption. The current traces from the patch-clamp experiments indicate that, as seen with the mature pores, Smp24 does not create ordered peptide-pore assemblies which would yield consistent and repeatable conductance levels. Instead, the peptide produces a range of different event types which often both have a variable conductance level within each individual event and even within the same event-type category. This indicates that the structure and size of the pores corresponding to the individual events also ranges widely and therefore the molecular-level structures of the peptide-pore assembly is likely disordered in its nature. The three different conductance-event types observed also indicate that multiple different biophysical mechanisms underly the membrane disruption at these early stages of the activity rather than just toroidal pores as observed for the mature pores. This observation that a variety of different event types can be seen for a single peptide is not uncommon for AMPs. While some examples of pore formation result in consistent and distinct conductance levels leading to square-top or flickering current signatures [33,34], they are not always consistent between different studies [35]. Many AMPs show either erratic behaviour or a combination of different event types like Smp24 [35,36,37,38,39]. One particularly interesting example is the cyclic beta sheet AMP gramicidin S. While having very little in common with Smp24 both structurally and in origin, the conductance behaviour of the two peptides is remarkably similar [40]. Due to the structural dissimilarity between the two peptides, this again indicates that the disruption occurs in a more disordered manner rather than by the formation of distinct structural assemblies. The multitude of different event types does not correlate well with the three traditional models for pore formation. However, alternative models such as the SMART model put a greater emphasis on the mechanism of action being not only dependent on the structure of the peptide but also the other conditions of the specific system such as the peptide concentration and the properties of the bilayer [5]. Such models better account for an individual peptide acting via multiple competing mechanisms of action, depending on the local conditions during a specific membrane-disruption event.

However, even if disordered, there must still be some correlation between the structure of Smp24 and the range of molecular-level behaviours leading to the occurrence of the conductance events. The distinct start and endpoints of the spike and multilevel events suggest that these events are pore-like in nature and must somehow be induced and stabilised by the presence of the peptide in the bilayer. In order to formulate more accurate models for these individual categories of membrane disruption, the specific structure of Smp24 and how it correlates with the orientation of the peptide must be considered, but the transient changes in the peptide orientations necessary to explain these very short-lived, early-stage pore forming events cannot readily be investigated using traditional experimental techniques. However, with a nanosecond time scale and atomic resolution, MD simulations can at least provide a theoretically based prediction as to how the peptide might behave under conditions mimicking those observed in the patch-clamp experiments. While the simulation of spontaneous AMP-induced pore formation is currently not feasible, the behaviour of the peptides in a pore-like environment can still be investigated, such as by manually inducing the formation of a toroidal using electroporation. This approach is especially useful for investigating the structure, orientation and position of peptides relative to the pore lumen, but due to the artificial conditions of the pore creation and stabilisation, it does not yield strong predictions with regard to more macroscopic pore-related factors such as the overall pore diameter or ideal number of peptide monomers in the pore assembly. Still, understanding how the behaviour of individual peptides changes between before and after pore nucleation and relative to the total concentration of peptides within the bilayer, could allow for a better contextualization of the experimental observations and the construction of more accurate models explaining the underlying structures of the different conductance events.

### 4.2. MD Simulation-Derived Structure of Smp24

Before the pore-associated structures and orientations of the peptide can be explored, the baseline peptide behaviour must be established. The configuration of the peptide inserted into the leaflet of a bilayer at a low peptide-to-lipid ratio was simulated, in conditions known to yield consistent structures for amphiphilic AMPs [41]. The simulated structure of Smp24 shows that the peptide has a number of characteristics that are consistent with many other peptides of the same class, but some aspects are also unique. The presence of a large helical region orientated in parallel with the bilayer is a well-known and experientially consistent structural characteristic of many AMPs [12,41,42,43]. In addition, as for Smp24, the presence of a proline residue has been shown several times to introduce a kink in the helical region although in most cases the kink is positioned more centrally in the helical region of the peptide [9,10,11,12]. However, the presence of the large unstructured region is a less common structural motif for AMPs of a similar size. The closely related peptide pandinin 2 also has an unstructured region near the C-terminal consisting of the same four residues as Smp24 (although ordered differently); however, unlike Smp24, this region is not separated from the rest of the peptide by the inclusion of a linker region [44]. The only other peptide found with a similar potential linker motif is another scorpion venom-derived peptide Con10. This peptide also contains three glycine residues near the C-terminal followed by a region of only polar or charged residues, which could give this peptide a similar overall 3D structure to Smp24. However, no detailed structural information has been published for this peptide yet [45,46]. The presence of the glycine linker region in Smp24 could affect the functionality of the highly polar tail region as the linker ensures that the last four residues are freely positioned around the lipid headgroups without affecting the orientation and positioning of the helical parts of the peptide.

### 4.3. Molecular Mechanisms and Structure behind Early-Stage Smp24-Induced Pores

With a baseline established for the structure of peptide and bilayer in simple conditions, further investigation can be carried out into the interplay between them and how their structures adapt as the conditions near pore formation. AMPs such magainin 2 and melittin have previously been shown to affect bilayers in non-disruptive ways prior to pore formation such as by inducing membrane thinning [47,48]. Smp24 itself has been shown to affect bilayer properties in several non-destructive ways, from introducing thinning defects to affecting the lipid ordering of phase separated bilayers [14]. The simulations show that Smp24 affects both the area per lipid, membrane thickness and the lipid chain order in a concentration-dependent manner. These changes are all likely related and stem from the way the top leaflet adapts to the presence of the inserted peptides. As the peptides take up space among the lipid phosphate and glycerol groups, these parts of the lipids get forced apart leading to the increase in the area per lipid. As the lipids are more spread out some lipid chains need to bend below the peptide in order to retain the integrity of the hydrophobic core of the bilayer, which in turn causes a reduction in the lipid chain order of the lower half of the chains. As an increasing number of the lipid chains in the top leaflet are no longer straight, a thinning effect is seen. These subtle concentration-dependent changes to the bilayer properties could play a part in reaching conditions that allow for further and more drastic membrane disruption.

#### 4.3.1. Concentration-Dependent Membrane Disruption

Both the patch-clamp experiments and the peptide-pore simulations indicate that membrane disruption is concentration dependent. The kinetic data from the patch-clamp experiments suggest that pore formation is a stochastic process that increases in probability and severity at concentrations above 3.8 µM. At the two lowest concentrations tested, the variations in the kinetics were very large with some instances where the disruption kinetics were in line with what was observed at the higher concentrations and some cases where no disruption was observed at all. Under these experimental conditions, pore formation is highly dependent on the local environment of the bilayer. Slight defects in the bilayer or small variations in the other experimental conditions may be enough to make the difference between complete destruction of the bilayer and no effect at all, while the behaviour is more consistent as the peptide concentration increases. At low peptide-to-lipid ratios, the MD simulations suggest that the peptides do not spontaneously transition into a configuration deep within the pore lumen. Instead, the most favourable peptide configurations seem to be when the peptide is associating with the interface between the pore and the bilayer, as this facilitates a better alignment between the curvature of the bilayer and the curvature of the helical region of the peptide induced by the proline14 residue. This principle is also demonstrated by the configuration with the secondary helix orientated towards the pore lumen (Figure 5B) being the most common, as this places the kink in a higher curvature region of the pore and allows the overall shape of the peptide and pore to better align compared with the reverse orientation. However, as the peptide-to-lipid ratio used in the simulation was increased, the peptides gained an ability to adopt more extreme configurations within the pore lumen. This is not due to direct interactions between the individual peptides but because of their concentration-dependent effects on the properties of the bilayer as a whole. As more peptides are inserted into the top leaflet of the membrane, a mismatch occurs between the ideal size of the two leaflets, which facilitates the net movement of lipid towards the bottom leaflet. As the peptides are directly interacting with the lipids, they can be pulled down into the pore lumen as the lipids are translocating. Furthermore, the lower lateral-diffusion constant of the peptides at the high peptide-to-lipid ratio can make it more difficult for the peptides to move out of the pore lumen. Together, this allows the peptides to adopt these more extreme configurations, even if the alignment between the helical and bilayer curvatures is less favourable. However, deeper positioning within the pore would likely provide an increase in the stabilising effect that the peptides have on the overall structure and the lifetime of the pore compared with the earlier configurations.

This molecular-level mechanism of a concentration-dependent change to the peptide-pore structure correlates well with the observations from the patch-clamp experiments. As the transition of the peptide to the most pore-stabilising configurations does not happen spontaneously but is very dependent on the specific local conditions around the area of the pore, one would expect to see a large degree of variation in the kinetics of the membrane disruption, especially at lower peptide concentrations. Furthermore, as deeply inserted peptide configurations are not intrinsically more energetically favourable, the nature of the pore structure would be expected to be disordered, leading to an inconsistent conductance level.

#### 4.3.2. Molecular-Level Structures Corresponding to Disruption Events

The behaviours found in the MD simulations allow us to propose different molecular-level structures corresponding to the different event types observed in the patch-clamp experiments (Figure 8). The distinct and clear beginning and end to the signals corresponding to the multilevel and spike events both suggest that these events are mechanistically related and that they correspond to a pore which has a distinct open and closed state, even if the characteristics of the open state are variable. The main difference is the lifetime of the open state, which, on a molecular level, corresponds to how well the pore is stabilised by the incorporation of the peptides. During both types of events, the local conditions of the bilayer are such that a pore-nucleation event can occur, either directly due to structural disruptions due to the presence of the peptides or indirectly via the macroscopic effects the inserted peptides have on the bilayer. In the case of the multilevel events, a number of peptides will transition from the pore interface into variable positions within the pore lumen providing stability to the pore structure, increasing the lifetime of the event. However, for spike events, this transition does not occur either due to chance or because the local peptide concentration around the pore is too low. As such, the pore will quickly close due to the line tension of the bilayer. The structural models for these peptide-induced pores could thus be best described as supported or unsupported disordered toroidal pores (Figure 8A,B). Proposing an underlying molecular structure corresponding to the erratic event type is more challenging. The lack of a distinct start and endpoint makes it less likely that the “pore” structure is similar to a water channel in the traditional sense. Instead, the event type could represent a more general form of disruption of the membrane structure increasing the “leakiness” of the bilayer, allowing for water and ions to sporadically cross the bilayer at an increased rate. Changes to the structure of the bilayer such as membrane thinning, increased lipid chain disorder, a higher portion of membrane defects or lipid removal via micelle formation could all be factors lowering the intrinsic resistance of the bilayer and thereby lead to periods of increased conductance (Figure 8C). Another option could be the formation of less well-defined peptide–lipid aggregates. It has previously been proposed that micellar-like peptide–lipid aggregates could form within the bilayer creating structures that function as a pore but without the toroidal or channel-like shape [49]. This would produce a much less defined path for the water and ions to penetrate the bilayer which could explain the more gradual shifts in the conductance seen for these events (Figure 8D).

## 5. Conclusions

The mechanism of action of AMPs such as Smp24 is varied and complex, especially during the early stages of membrane disruption. For Smp24, patch-clamp experiments indicate that multiple mechanisms of pore formation can be present at the same time, something which is not well accounted for with the ridged-pore structures proposed in the three traditional models for AMP-induced pore formation. Instead, a more general understanding of the ability for both the peptide and bilayer to transiently respond and adapt their structures to the changing conditions that occur after the peptides and the bilayer meet is required to explain this phenomenon. A combination of the off-centre kinked structure of the peptide and the concentration-dependent effect of an uneven distribution of the peptide across the bilayer leaflets can explain why some short-lived disordered toroidal pores transition into more stable events while others do not.

## Figures and Tables

**Figure 1 pharmaceutics-15-02399-f001:**
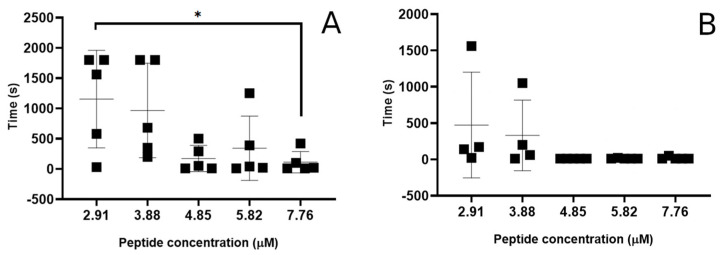
Smp24 pore formation kinetics via patch clamp. (**A**) Time between addition of peptide to the bilayer and the occurrence of an irreversible disruption of the bilayer resistance. (**B**) Time between the addition of peptide to the bilayer and the observation of the first conductance event. Data shown are mean and standard deviation (*n* = 5), * indicates significant difference based on unpaired *t*-test (*p* < 0.05).

**Figure 2 pharmaceutics-15-02399-f002:**
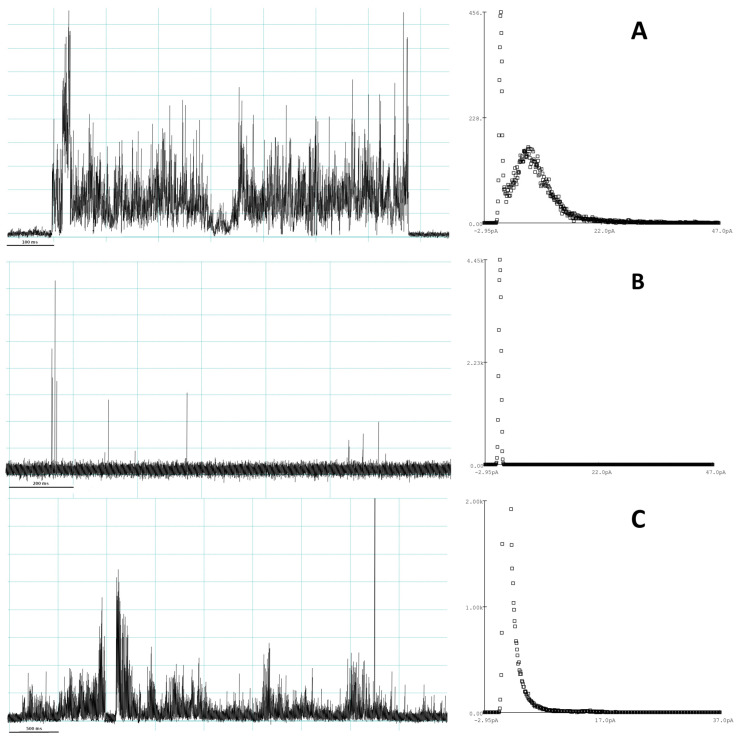
Representative examples of the current trace and amplitude histograms for the three different event types observed in the patch-clamp experiments. (**A**) Multilevel events, (**B**) spike events, (**C**) erratic events.

**Figure 3 pharmaceutics-15-02399-f003:**
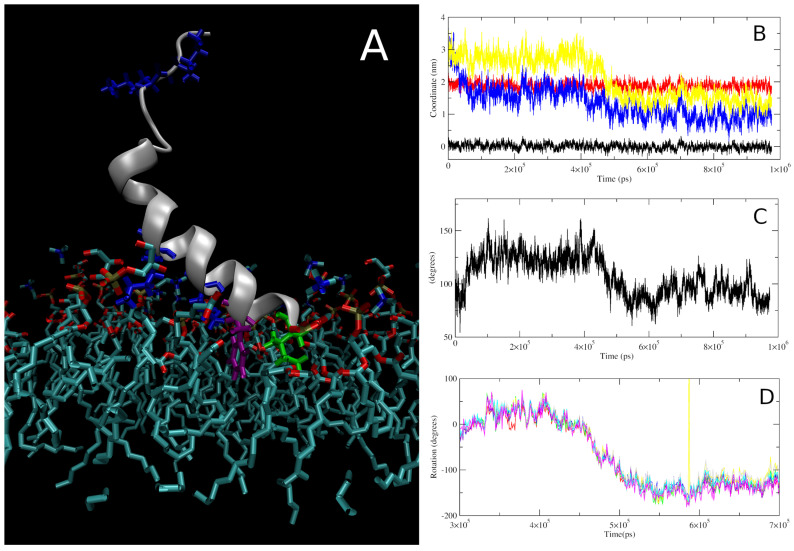
Characteristics of the insertion of Smp24 into the negative bilayer. (**A**) Configuration of the peptide in the N-terminal anchored stage of the insertion. Interactions occur via two of the four lysine residues (blue), the N-terminal isoleucine (green) and the position 4 phenylalanine (purple). (**B**) Changes over time in the Z-axis centre of mass of peptide (yellow) and N-terminal (blue) relative to the phosphor atoms of the top leaflet (red) and centre of the bilayer (black). (**C**) Changes over time in the tilt angle relative to the bilayer norm of the helical region from residues 1–12. (**D**) Cumulative changes in the local helical rotation of residue 2–10 during the rotational stage of the insertion process. Figures shown are based on one simulation; corresponding figures for all repeats can be found in Appendix A.

**Figure 4 pharmaceutics-15-02399-f004:**
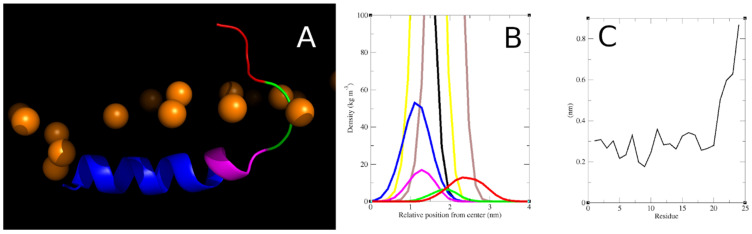
(**A**) Three-dimensional structure of Smp24 inserted into a PCPG bilayer. Blue = primary helix (r1–13), magenta = secondary helix (r14–17), green = glycine linker (r18–20), red = tail (r21–24), orange = lipid phosphor atoms. (**B**) Partial density profiles of Smp24 inserted into a PCPG bilayer, with positions relative to the centre of the bilayer. Black = lipid acyl chains, yellow = lipid glycerol esters, brown = lipid headgroups with phosphates; peptide regions/colours are the same as in (**A**). (**C**) Per residue RMSF of Smp24 after insertion into the negative bilayer. Figures shown are based on one simulation, corresponding figures for all repeats can be found in Appendix A.

**Figure 5 pharmaceutics-15-02399-f005:**
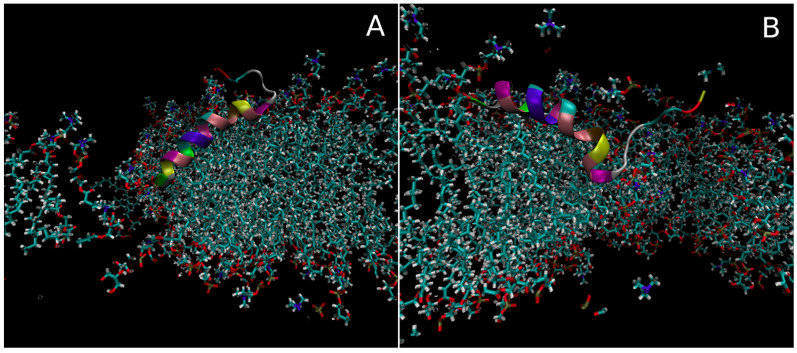
Three-dimensional models of the two pore-associated configurations seen for Smp24 in the long_pbcg_1–3 simulations. (**A**) The helical regions of Smp24 are aligned with the curvature of the pore interface with the secondary helix positioned within the top of the pore. (**B**) The helical regions are aligned in reverse such that the primary helix is positioned within the top half of the pore.

**Figure 6 pharmaceutics-15-02399-f006:**
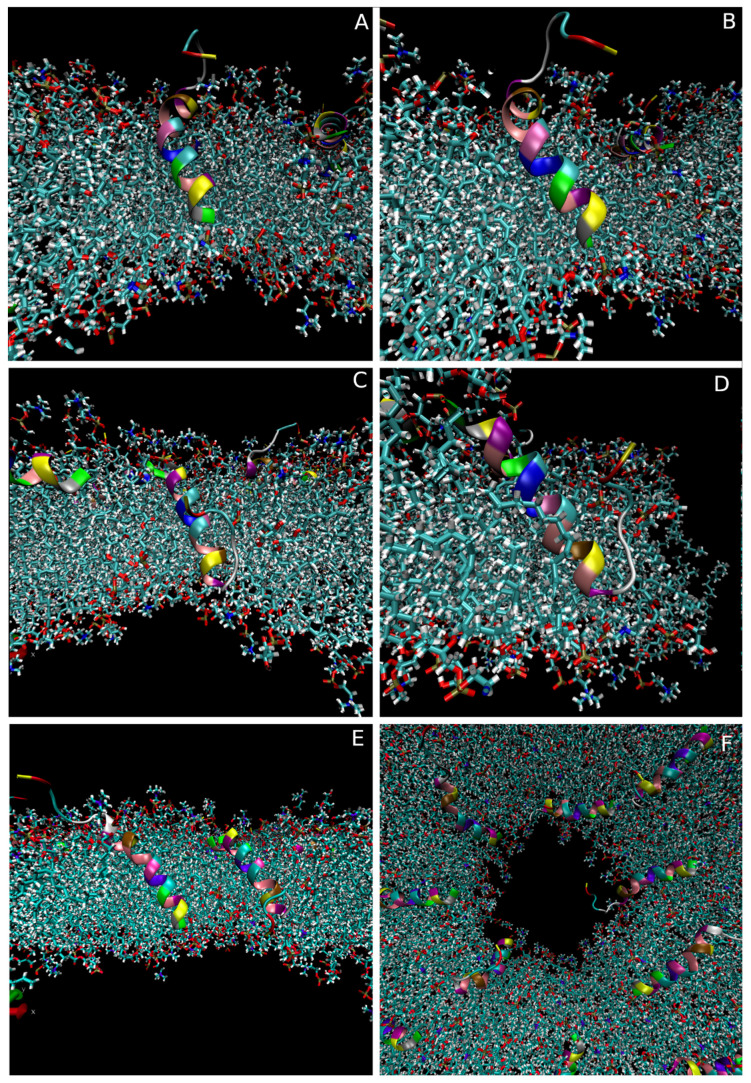
Examples of transmembrane peptide configurations in the multi-peptide pore simulations. (**A**,**B**) Example of peptide with the primary helical region positioned the deepest in the pore lumen (frontal and sideways view), (**C**,**D**) example of peptide with the secondary helix positioned the deepest in the pore lumen (frontal and sideways views), (**E**) two peptides in transmembrane configurations in the same pore, (**F**) example of a top-down view of a pore with multiple peptides associated with the pore interface.

**Figure 7 pharmaceutics-15-02399-f007:**
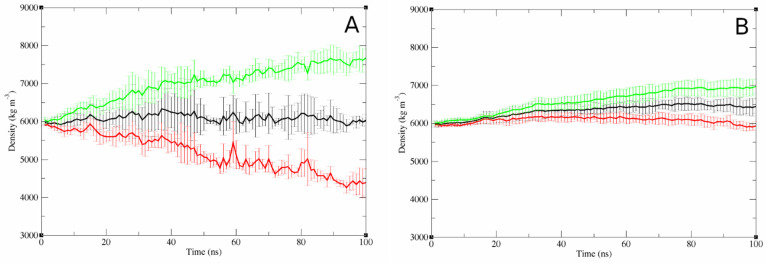
Translocation of lipids through the membrane pore during the simulations. Results are shown as the average ± SD lipid density in the bottom leaflet over time for the pore simulations. (**A**) Single peptide pore models (*n* = 3), (**B**) multi-peptide pore models (*n* = 5). Black = combined density for both DOPC and DOPG lipids divided by 2, red = density of DOPC lipids, green = density of DOPG lipids.

**Figure 8 pharmaceutics-15-02399-f008:**
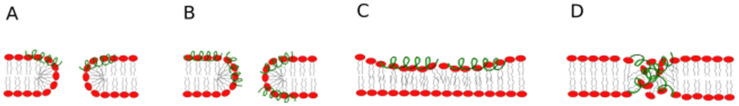
Proposed molecular-level structures corresponding to the conductance events observed in the patch-clamp experiments. (**A**) The short-lived spike events are caused by unsupported toroidal pores without peptides in the pore lumen. (**B**) The longer-lived multilevel events are caused by supported toroidal pores with peptides taking part in the pore structure. (**C**) General disruptions to the bilayer structure such as membrane thinning could be the reason for the erratic events. (**D**) Micellar-like aggregates within the bilayer could also be the reason for erratic events.

**Table 1 pharmaceutics-15-02399-t001:** Concentration-dependent effects of peptide insertion on the structure and order of the lipid bilayers. Values represents the average ± the standard deviation (SD) over the simulation period.

Number of Peptides	Peptide-to-Lipid Ratio	Bilayer Size (nm^2^)	Area per Lipid (nm^2^)	Bilayer Thickness (nm)	Average Shift in the Lipid Order Parameters of C13–18 Relative to the Bilayer-Only Simulation (Scd)	Lateral Peptide Diffusion Constant (10^−7^ cm^2^/s)
0	na	13.64	0.696 ± 0.006	3.797 ± 0.029	na	na
4	1:144	14.21	0.707 ± 0.006	3.789 ± 0.030	2.9 × 10^−4^ ± 6.85 × 10^−4^	1.52 ± 0.52
9	1:48	12.62	0.736 ± 0.007	3.752 ± 0.030	−5.36 × 10^−3^ ± 1.51 × 10^−3^	0.99 ± 0.37
16	1:32	14.20	0.758 ± 0.006	3.711 ± 0.029	−1.02 × 10^−2^ ± 1.53 × 10^−3^	0.32 ± 0.26

## Data Availability

Not applicable.

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
