# Peer review of "Mechanistic Insight into the Early Stages of Toroidal Pore Formation by the Antimicrobial Peptide Smp24"

_pharmaceutics, 2023, doi:10.3390/pharmaceutics15102399_

Round 1

Reviewer 1 Report (Previous Reviewer 3)

Having reviewed the authors' point-by-point response to my comments, I believe that the manuscript should be accepted in its current form.

Additionally, it's important to note that the figures within the paper suffer from low image quality, and the font size of the axis values is insufficient, making it difficult to discern details in the figures.

Author Response

Response to reviewers comments for submission pharmaceutics-2577200

Dear editor,

We thank the reviewers for their comments and have the following responses: -

Reviewer 1 –

Having reviewed the authors' point-by-point response to my comments, I believe that the manuscript should be accepted in its current form.

Additionally, it's important to note that the figures within the paper suffer from low image quality, and the font size of the axis values is insufficient, making it difficult to discern details in the figures.

We are happy to follow any request from the journal editorial team regarding modification of the figures.

Yours sincerely,

Keith Miller.

k.miller@shu.ac.uk

Reviewer 2 Report (New Reviewer)

The authors report lab bench biochemical experiments to investigate the early stages of toroidal pore formation by the antimicrobial peptide Smp24. They also report molecular dynamics simulations to investigate the peptide structure and orientation before and during pore formation at the molecular level.

Line 133 ‑ I could not find any reference to previous peptide-membrane models via electroporation. I advise the authors to write a section describing the setup of the electroporation simulation or else place it in the Supplementary Material since I could not find a description in the Materials and Methods section.

Lines 137-138 – The authors must show that the MD simulations are well equilibrated; therefore, it is necessary to include the energy convergence plots of the following experiments: three repeats, 500 ns in length, and five repeats, 100 ns in length.

Line 238 ‑ The authors must place the correct values in the abscissa in plots 3B, 3C, and 3D. These values do not correspond to the time described in line 236 regarding the observed time of 50 ns when the peptide reached a stable orientation. Note that 5e+05 picoseconds are equal to 500 nanoseconds, not 50 ns, as stated.

Line 247 ‑ There is no figure S1 in the Supplementary Material.

Line 283 – To define the acronym SCD, I suggest modification of the phrase "deuterium order parameters of the lipid chains" to "deuterium order parameters of the lipid chains (SCD)". The authors must show plots of the local lipid chain order parameter for the POPC and POPG lipids.

Line 284 ‑ There is no figure S3 in the Supplementary Material.

Line 307 ‑ The lack of transmembrane peptide insertion might need larger simulation times or a larger number of monomers.

Line 325 ‑ Single peptide? - Three continuing simulations or three replicas?

Lines 370-371 ‑ There is no discussion of figures 6C and 6D in the text. Discussing the peptide arrangement of the Smp24 peptides forming a pore is most important. Lateral and side views are needed. It is unclear how the distribution shown in Figure 6D can induce pore formation. No information is given about the pore diameter, nor is a discussion presented regarding the ions that could pass through this pore. The authors should use g_lomepro; Gapsys, V., de Groot, B. L., & Briones, R. (2013) to better illustrate the membrane properties. Computational analysis of local membrane properties. Journal of computer-aided molecular design, 27, 845-858. http://www3.mpibpc.mpg.de/groups/de_groot/g_lomepro.html

Line 530 ‑ The discussion of the membrane disruption must include an analysis of the data obtained after applying g_lomepro software.

Author Response

Response to reviewers comments for submission pharmaceutics-2577200

Dear editor,

We thank the reviewers for their comments and have the following responses: -

Reviewer 2 – 

Line 133 ‑ I could not find any reference to previous peptide-membrane models via electroporation. I advise the authors to write a section describing the setup of the electroporation simulation or else place it in the Supplementary Material since I could not find a description in the Materials and Methods section.

More detailed protocol added to supplementary materials  

Lines 137-138 – The authors must show that the MD simulations are well equilibrated; therefore, it is necessary to include the energy convergence plots of the following experiments: three repeats, 500 ns in length, and five repeats, 100 ns in length.

Potential energy convergence plots added to supplementary materials  

Line 238 ‑ The authors must place the correct values in the abscissa in plots 3B, 3C, and 3D. These values do not correspond to the time described in line 236 regarding the observed time of 50 ns when the peptide reached a stable orientation. Note that 5e+05 picoseconds are equal to 500 nanoseconds, not 50 ns, as stated.

Clarified in main text which period is referred to on the corresponding figures  

Line 247 ‑ There is no figure S1 in the Supplementary Material.

Figure names in supplementary materials updated  

Line 283 – To define the acronym SCD, I suggest modification of the phrase "deuterium order parameters of the lipid chains" to "deuterium order parameters of the lipid chains (SCD)". The authors must show plots of the local lipid chain order parameter for the POPC and POPG lipids.

Main text updated, figures for the individual lipid types added to supplementary materials.  

Line 284 ‑ There is no figure S3 in the Supplementary Material.

Figure names in supplementary materials updated

Line 307 ‑ The lack of transmembrane peptide insertion might need larger simulation times or a larger number of monomers.

Yes, we agree. A larger number of monomers seems to increase the likelihood of the peptides to adopt a transmembrane configuration, which could explain the threshold-based relationship between the pore formation and peptide concentration seen in the patch clamp kinetics in this study and in previously published biophysical experiments.

Line 325 ‑ Single peptide? - Three continuing simulations or three replicas?

Single peptide, three repeats, with independent formation of the pore in each simulation.
This has been further clarified in the text.

Lines 370-371 ‑ There is no discussion of figures 6C and 6D in the text. Discussing the peptide arrangement of the Smp24 peptides forming a pore is most important. Lateral and side views are needed. It is unclear how the distribution shown in Figure 6D can induce pore formation. No information is given about the pore diameter, nor is a discussion presented regarding the ions that could pass through this pore. The authors should use g_lomepro; Gapsys, V., de Groot, B. L., & Briones, R. (2013) to better illustrate the membrane properties. Computational analysis of local membrane properties. Journal of computer-aided molecular design, 27, 845-858. http://www3.mpibpc.mpg.de/groups/de_groot/g_lomepro.html

Sideways and frontal images of peptide configurations added.
All images referenced in main text.

 While we agree that investigating the arrangement of the peptides during the pore formation itself would be of great interest, simulating spontaneous AMP induced pore formation is not possible (see review by Lipkin et al,
http://dx.doi.org/10.1098/rstb.2016.0219). Therefore, as Lipkin suggests, the best we can do is investigate either how the peptides intact with pre-formed pores or manually insert the peptides in a formation similar to the expected pore structure. As the patch clamp experiments did not indicate that the pore structure was very ordered, we chose the first option for our MD simulations. We use electroporation to induce pores in the bilayers and this process occurs independently of the peptide (pore will also form in a bilayer without inserted peptides and we found no clear correlation between the location at which the pore formed relative to the location of the peptide (n=15)). The models therefore cannot say anything concretely about the formation of the pore itself, only how the peptide is likely to behave once it finds itself in a toroidal pore like environment and how that behaviour changes based on the number of peptides.  

While we do emphasize that we are looking at peptide-pore association in the text in the results section, the limitations of the models were not clearly enough communication in the discussion which we now have improved.

Furthermore, upon review there definitely where a few instances where the use of the words "pore formation" was applied in an imprecise manner (especially in subheadings) and these have now been changed to something more appropriate.

Approximate pore sizes are mentioned in the text, but more detailed investigation of the pore sizes is not done as it is likely also very dependent on other factors used in the modelling (bilayer size, strength of electric field ect) and not just the presence of the peptides in the pore lumen. Again, the scope of the pore models is mostly limited to contextualizing what orientations the peptides are likely to adopt in a pore like environment (and how it changes based on the peptide-lipid ratio) and will not necessarily give the most accurate overall pore-peptide assemblies in terms of pore size and number of monomers in taking part in the structure. Therefore, we want to be really careful to not read too much these aspects of the models.

Lastly, the experimental results (both the patch clamp data in this study and previously published AFM investigations) have shown that Smp24 induces pores without any specific sizes (that can expand to a diameter > 100 nm), so evaluation of whether specific ions can penetrate or not is not as relevant as with other peptides (such as alamethicin) that form more ordered pore structures.

We agree that some of the tools in the g_lomepro programme (such as better visualising of the local curvature of the pore) could potentially improve the visualisation of the peptide-pore alignment, however unfortunately the g_lomepro programme is rather old. It says on the github page that the pre-compiled version of g_lomepro should be compatible with newer versions of Gromacs but we have not been able to get it to work (tested on two different computers with two different versions of gromacs). In order to compile the programme manually requires a 10 years old version of gromacs (version 4.6 or older), which is no longer officially supported. We did manage to find a repository with gromacs4.6 but were unable to successfully compile this version on our modern hardware.

Line 530 ‑ The discussion of the membrane disruption must include an analysis of the data obtained after applying g_lomepro software

See previous answer.

Yours sincerely,

Keith Miller.

k.miller@shu.ac.uk

Reviewer 3 Report (New Reviewer)

The manuscript entitled: “Mechanistic insight into the early stages of toroidal pore for-2 mation by the antimicrobial peptide Smp24”, is a very interesting study of the mechanism of AMP (in this work a scorpion venom peptide) at the early stages of their interaction with cell membranes (using patch clamp and computational models). It is well written and reads very easy. The results are also clearly explained and well complemented by figures and tables as well as by supporting information.

Said this, the only comments that I have are:

- Since the authors seem to be English native speakers, this work is well written and has no typos or grammatical issues. Only three minor comments:

            - On lines 57-58, the “(Munich, Germany” names need to be closed by another parenthesis

            - Table 1 (starting on line 294) has the headlines and he whole data spread out. This is probably just a formatting issue on the PDF manuscript, but just to mention it.

            - Similar issue with the references at the end of the manuscript. The line spacing is wider than the rest of the text. Again, it can be a matter of text formatting.

In all, a nice work that I recommend for publication.

Author Response

Response to reviewers comments for submission pharmaceutics-2577200

Dear editor,

We thank the reviewers for their comments and have the following responses: -

Reviewer 3 –

The manuscript entitled: “Mechanistic insight into the early stages of toroidal pore for-2 mation by the antimicrobial peptide Smp24”, is a very interesting study of the mechanism of AMP (in this work a scorpion venom peptide) at the early stages of their interaction with cell membranes (using patch clamp and computational models). It is well written and reads very easy. The results are also clearly explained and well complemented by figures and tables as well as by supporting information.

 Said this, the only comments that I have are:

 - Since the authors seem to be English native speakers, this work is well written and has no typos or grammatical issues. Only three minor comments:

             - On lines 57-58, the “(Munich, Germany” names need to be closed by another parenthesis

 We have corrected this in the text.

            - Table 1 (starting on line 294) has the headlines and the whole data spread out. This is probably just a formatting issue on the PDF manuscript, but just to mention it.

This appears to be an issue with the pdf conversion, we have rechecked the original Word file and the spacing is fine. We will raise it with the journal editorial staff at the proof stage.

            - Similar issue with the references at the end of the manuscript. The line spacing is wider than the rest of the text. Again, it can be a matter of text formatting.

 Again, this appears to be a conversion issue we will double check at the proofing stage.

In all, a nice work that I recommend for publication.

Yours sincerely,

Keith Miller.

k.miller@shu.ac.uk

Reviewer 4 Report (New Reviewer)

At present, the emergence of multi-drug resistant (MDR) pathogens worldwide has presented unprecedented challenges in the clinical treatment of infections. In this scenario, it is imperative to develop novel antimicrobials, and the antimicrobial peptides have emerged as potential candidates and constitute an inexhaustible source of new antimicrobial molecules. In this article submitted by Magnus Bertelsen et al., and entitled "Mechanistic insight into the early stages of toroidal pore formation by the antimicrobial peptide Smp24", the author specifically investigates the critical early stages of antimicrobial peptide Smp24 induced membrane disruption using planar patch clamp Electrophysiology and molecular dynamics simulations. The paper is highly intriguing and innovative; however, there are certain questions that the author needs to address before its publication.

1. In the section of introduction the background information associated to “the antimicrobial peptide Smp24” should be provided. Otherwise, the sequence of Smp24 or detailed information should be presented.

2. The concentrations of "Smp24" in the Electrophysiology experiments should be explicitly stated in the section on "2.1. Patch clamp," based on factors such as MICs or other relevant criteria.

3. The description in this paragraph from line 145 to 152 should be moved to the "discussion" section.

4. The "Conclusions" section should be more succinct and emphasized.

5. It would be better if a mechanism diagram illustrating the primary actions of "antimicrobial peptide Smp24 induced membrane disruption" could be provided.

Author Response

Response to reviewers comments for submission pharmaceutics-2577200

Dear editor,

We thank the reviewers for their comments and have the following responses: -

Reviewer 4 –

1. In the section of introduction the background information associated to “the antimicrobial peptide Smp24” should be provided. Otherwise, the sequence of Smp24 or detailed information should be presented.

Peptide sequence added to introduction

2. The concentrations of "Smp24" in the Electrophysiology experiments should be explicitly stated in the section on "2.1. Patch clamp," based on factors such as MICs or other relevant criteria.

Concentrations added to the methods section.

The concentration range is somewhat based on trial and error as (especially in biophysical experiments using synthetic membranes) the activity level of the peptide is also based on the concentration of lipids. The concentration range used in this study fits such that the lower end is relatively close to concentrations used for the peptide in previously published biophysical experiments, while the max concentration (7.76 µM) is around the MIC of the peptide vs S. aureus (MIC = 6.2 µM).  

3. The description in this paragraph from line 145 to 152 should be moved to the "discussion" section.

While we do agree that in principle this section might be more correctly placed in the discussion section, the inclusion of this section (at this point) is based on previous reviewer feedback to the results section that it was unclear why we had chosen the specific lipid composition for the experiments. Thus, to avoid reader confusion we think that including this “mini-discussion” (that content wise is completely independent from the rest of the discussion and the results) before any experimental results are presented, improves the overall readability of the paper to a much greater extent than if placed later in the discussions section. 

4. The "Conclusions" section should be more succinct and emphasized.

Conclusions have been shortened.  

5. It would be better if a mechanism diagram illustrating the primary actions of "antimicrobial peptide Smp24 induced membrane disruption" could be provided.

We agree that illustrating the primary mechanism of action of Smp24 would be ideal, but we want to be careful not to make too strong conclusions based on the results of this study. The results are limited to only the early stages of the pore formation/stabilisation, which we still think is very important for understanding the structure activity relationship of the peptide but might not be directly translatable to the primary mechanism of membrane disruption in bacteria (especially since the investigations are done in a synthetic system.

Future work is still needed to explain how the small toroidal pores described in this study can grow to the much larger pores observed using AFM in previous studies (Harrison et al, Phospholipid dependent mechanism of smp24, an α-helical antimicrobial peptide from scorpion venom. Biochim Biophys Acta. 2016;1858(11):2737-2744. doi:10.1016/j.bbamem.2016.07.018) and ideally the presence of toroidal pores should also be demonstrated in a true bacterial system. 

Yours sincerely,

Keith Miller.

k.miller@shu.ac.uk

Round 2

Reviewer 4 Report (New Reviewer)

The article titled "Mechanistic insight into the early stages of toroidal pore formation by the antimicrobial peptide Smp24," submitted by Magnus G Bertelsen, et al., has been carefully revised by the authors in response to review comments. As a result, the significant improvements have been made to the manuscript. Therefore, I fully concur that this paper is suitable for acceptance in its current form.

.

This manuscript is a resubmission of an earlier submission. The following is a list of the peer review reports and author responses from that submission.

Round 1

Reviewer 1 Report

The article entitled "Mechanistic insight into the early stages of toroidal pore formation by the antimicrobial peptide Smp24" discusses the mechanism of action of the antimicrobial peptide Smp24 and its potential as a drug candidate. The article explores different models for peptide-pore structures that facilitate membrane disruption, including the traditional barrel stave, toroidal pore, and carpet models. The authors argue that a more general understanding of how peptides and bilayers respond and adapt to changing conditions is necessary to fully explain the diversity seen in membrane disruptive behavior. Overall, this report lacks in vitro trials to illustrate the biological mechanism and therapeutic potential of Smp24. In addition, the format of the data and graphs is extremely irregular. Molecular dynamics data with a length of 100 ns cannot draw conclusions about pore formation models.

Moderate editing of English language required

Reviewer 2 Report

The manuscript from Bertelsen and collaborators provides more insights into the structure-function relationship of the antimicrobial peptide Smp24. 

The study was well designed and presented, thus it deserves be published after minor adjustments.

1) Abstract

- The aim of the study is not properly presented in the abstract;

- Keywords are not provided.

2) Introduction 

Line 55: please provide the meaning of each abbreviation at the first citation, such as AFM, QCM-D.

3) Methods 

- Please provide a ‘Statistical analysis’ section describing how the data analysis was carried out.

4) Results 

- The authors should improve the presentation of table 1. It should be formatted following the journal rules. 

- Table 1: The title of table 1 should be adjusted to make it more clear and informative.

- Figure 6: The title of this figure should be adjusted to make it more clear and informative. 

5) Discussion 

- This section is usually presente without subsections. The authors should consider this modification. 

Reviewer 3 Report

In the present study, Bertelsen et. have studied the early stages of toroidal pore formation by the antimicrobial peptide Smp24. There are serious technical issues with the manuscript in addition to that results are not also well presented please find my comments below:

1) Why the authors have chosen DOPG: DOPC membrane in these simulations, and why not other lipids?  Why authors have chosen compositions of 50:50 for the membrane? authors need to justify in detail.

2) Why did the authors not equilibrate the membrane before performing the peptide membrane complex?  Since the membrane is not equilibrated its highly likely chances it may not behave correctly during the simulations.

3) Why simulations were performed for different lengths?

4) Why do authors perform simulations at two different temperatures?

5) Also, results are not presented very well e.g. time evolution of Z-axe distances must be included in the main manuscript, moreover, the z-axis distances should be calculated between COM of peptide and membrane. 

Reviewer 4 Report

This manuscript studies the disruption of bacterial membrane by antimicrobial peptide Smp24. They explored the mechanism in the early stage of this procedure by both patch clamp and molecular dynamics simulations. Their findings could provide more information for explaining the interaction between bacterial membrane and antimicrobial peptides.

Major questions:

1. In the patch clamp experiment and simulations, they used a simply lipid planar bilayer. How did this planar lipid bilayer model represent the properties of bacterial membrane for both gram positive and negative strains? Does the outer membrane of gram negative strains or peptidoglycan could affect the results of experiments and simulation? 

2. It seems the MD simulations only tested the first two events observed in the patch clamp experiments (Figure 7 A&B). And these two events had a pre-established pore in the lipids membrane before meet peptide. Could authors also do the simulation for other two events (Figure 7C&D)?

Minor comments:

1. Please explain what DOPC and DOPG lipids are.

2.  Figure 1A, there seems be only four data points in the two left conditions, could author double-check it?